# Prevalence of Nasal Shedding of Equid Gammaherpesviruses in Healthy Swiss Horses

**DOI:** 10.3390/v13091686

**Published:** 2021-08-25

**Authors:** Laura Scheurer, Claudia Bachofen, Isabelle Hardmeier, Julia Lechmann, Angelika Schoster

**Affiliations:** 1Klinik für Pferdemedizin, Departement für Pferde, Vetsuisse Fakultät Zürich, Winterthurerstrasse 260, 8057 Zürich, Switzerland; laura.scheurer@uzh.ch (L.S.); isabelle.hardmeier@uzh.ch (I.H.); 2Institut für Virologie, Vetsuisse Fakultät Zürich, Winterthurerstrasse 266a, 8057 Zürich, Switzerland; cbacho@vetvir.uzh.ch (C.B.); julia.lechmann@uzh.ch (J.L.)

**Keywords:** gamma herpes virus, EHV-2, EHV-5, AHV-5, horse, viral shedding

## Abstract

Equid Gamma herpesvirus (eGHV) infections have been reported worldwide and may be correlated with clinical signs, e.g., affecting the respiratory tract in young horses. eGHV are shed by healthy horses as well as horses with respiratory tract disease. The prevalence in healthy Swiss horses is unknown to date but this data would provide valuable information for causal diagnosis in clinical cases and formulation of biosecurity recommendations. Nasal swabs from 68 healthy horses from 12 Swiss stables and 2 stables near the Swiss border region in Germany were analyzed by panherpes nested PCR. Positive samples were sequenced. A multivariable model was used to determine if sex, age, breed, canton, or stable had a significant effect on the shedding status of each detected eGHV. Overall, the eGHV prevalence was 59% (*n* = 68); the prevalence for equid herpesvirus-2 (EHV-2), equid herpesvirus-5 (EHV-5) and asinine herpesvirus-5 (AHV-5) was 38%, 12% and 9%, respectively. Co-infections with multiple eGHVs were observed in 25% of the positive samples. The odds of shedding EHV-2 decreased with age (*p* = 0.01) whereas the odds of shedding AHV-5 increased with age (*p* = 0.04). Breed, sex, canton, or stable had no significant association with eGHV shedding. As EHV-2 shedding was common in healthy horses a positive PCR result must be interpreted with caution regarding the formulation of biosecurity recommendations and causal diagnosis. As EHV-5 and AHV-5 shedding was less common than EHV-2, a positive test result is more likely to be of clinical relevance. Shedding of multiple eGHV complicates the interpretation of positive test results in a horse.

## 1. Introduction

Equid herpes viruses (EHV) are important pathogens with a worldwide prevalence [1,2,3,4]. The hallmark of herpes viruses is their ability to establish latency, during which no infectious virions are produced, and the virus escapes the immune system. Horses usually become infected with EHV early in life and in the case of alpha herpesvirinae, latency is established in regional ganglia and to some extent in leukocytes, while gamma herpesvirinae are thought to establish latency in the lymphatic system only. The virus may then be reactivated periodically throughout life during periods of immune suppression. During reactivation, infectious virus particles are shed via site-specific secretions with or without accompanying clinical signs, and virus shedding horses serve as a source of infection for other animals [5,6].

There are two subfamilies of herpes viruses with importance in equine medicine, alphaherpesviruses (e.g., EHV-1, EHV-3, and EHV-4) and equid gammaherpesviruses (eGHV; e.g., EHV-2, EHV-5, and EHV 7 (=AHV-2); Figure 1). Alpha herpes viruses are well studied, but less is known about eGHV.

The eGHV has longer replication cycles than alphaherpesviruses and establishes latency in B- and T-lymphocytes, in contrast to alphaherpesviruses, which prefer latency in the trigeminal ganglia and sensory neurons [5,7]. eGHV may cause respiratory tract disease in young animals [8,9]. The pathogenicity of EHV-2 and EHV-5 in adult equines is controversial, as both viruses can be recovered from the nasal secretions of clinically affected and healthy animals [4,10,11]. Exercise intolerance and immune suppression in adult animals have been linked to eGHV infection [12,13]. Recently, EHV-5 and EHV-2 have also been associated with equine multinodular pulmonary fibrosis (EMPF), chronic interstitial pneumonia of adult horses [14,15,16]. Experimental infections with EHV-2 and EHV-5 in vivo and in vitro have provided further evidence for eGHV involvement in pulmonary fibrosis [12,13,17]. However, it is unknown why some animals develop pulmonary fibrosis while others do not [18]. Little is known about the clinical importance of AHV-5 in horses other than being isolated from a horse with pyogranulomatous pneumonia and horses with EMPF [16,19].

EHV-2 is the most frequently detected eGHV in horses. Reported prevalence of EHV-2 in nasal swabs and peripheral blood mononuclear cells (PBMC) ranges from 0–93% [20,21,22,23,24] and 30–100% [8,20,25,26], respectively, depending on geographic location, age, and the investigated specimens. Reported EHV-5 prevalence is within a similar range, between 14 to 92% [11,20,21,24,27] and 0–88% [8,11,25,26] in nasal swabs and PBMC, respectively. In both cases, the highest prevalence is reported in young horses (<3 years); [24,25,26,27]. The prevalence of nasal shedding of eGHV in healthy horses in Switzerland is unknown. The clinical relevance of a positive eGHV PCR result in a nasal swab from a horse with signs of respiratory tract disease is difficult to interpret, as healthy animals can also shed the virus. Data on the local prevalence of virus shedding in healthy horses could help with the interpretation of such diagnostic results. A positive (nasal swab) PCR result for EHV-2 and EHV-5 also raises the question of whether the affected animal should be isolated from other horses. Data on nasal shedding in healthy horses in Switzerland may therefore help with recommendations regarding biosecurity measures and causal diagnosis of clinical cases.

The objective of this study was to determine the prevalence of eGHV shedding in nasal secretions of healthy horses in Switzerland.

## 2. Materials and Methods

### 2.1. Study Population

Out of a total of 200 nasal swabs collected for another study from healthy horses in 2017 (material intended for publication) [28], 68 archived DNA samples were randomly selected. The study for which the samples were collected was approved by the Swiss cantonal authorities for animal use (ZH216/16). DNA had been extracted from the nasal swabs using a commercial DNA extraction kit (Qiamp DNA mini kit, Qiagen, Hombrechtikon, Switzerland) according to the manufacturer’s instructions. DNA was stored at −80°C and had only undergone one thawing cycle before use in this project.

### 2.2. Panherpes Nested PCR

A panherpes nested PCR was performed as previously described [29,30]. This PCR targets a highly conserved region of the herpesviral DNA polymerase gene. As degenerate consensus primers were used, the PCR was capable of detecting known and novel herpesviruses without prior information of the DNA sequence [29,31]. The expected size of the product after panherpes nested PCR was 215–235 bp [30]. A minor modification to the reaction mixes described by Ehlers et al. (1999) was made [30]. In the first PCR round, 5 µL of extracted sample DNA was used and for the second PCR round, 1 µL of the product from the first round was applied. The final volume of both mixtures added up to 25 µL and contained 2.5 µL PCR Buffer (10×, Qiagen, Hombrechtikon, Switzerland), 200 µM of each deoxynucleotide triphosphate, 1µM of each primer, and 2 units HotStarTaq DNA Polymerase (5 U/µL, Qiagen, Hombrechtikon, Switzerland). DEPC treated water was added as needed to reach the final reaction volume of 25 µL. DEPC treated water was used as negative control and DNA extracted from leukocytes of a cow with confirmed bovine malignant catarrhal fever caused by ovine herpesvirus-2 was used as a positive control. The Peltier Thermal Cycler-200 (MJ Research) was used for thermal cycling according to the protocol of Ehlers et al. (1999) with a minor modification [30]. The initial denaturation time was 12 min at 95 °C instead of 3 min at 95 °C. Before loading the products from the second PCR on a 2% agarose gel, containing Gel Red (1000×, Biotium, Hayward, CA, USA), 5 µL of Orange Loading Dye (6×, Thermo Fisher Scientific, Waltham, MA, USA) was added. A 50 bp DNA Ladder (New England Biolabs, Ipswich, MA, USA) was used. Visualization of bands with the expected size (215–235 bp) was performed under UV light. Bands of the correct size were cut from the gel with a sterile scalpel blade. DNA was extracted from the Gel using the QIAquick^®^ Gel Extraction Kit according to the manufacturer’s instructions (Qiagen, Hombrechtikon, Switzerland). Elution of the DNA was performed with 30 µL of elution buffer, followed by the third PCR applying non-degenerated primers [29].

The reaction mixture for the third PCR was identical to the second PCR with exception of using only 200 µM of each sequencing primer and 1 unit of HotStarTaq DNA Polymerase (5 U/µL). The QIAquick^®^ PCR Purification Kit (Qiagen, Hombrechtikon, Switzerland) was used to purify the DNA. After measuring the total DNA concentration using Nanodrop (ND-1000 Spectrophotometer, FisherScientific AG, Kloten, Switzerland) samples were sent for unidirectional Sanger sequencing (economy run) to Microsynth GmbH (Balgach, Switzerland). The resulting sequences were compared to published sequences using NCBI BLAST^®^ (https://blast.ncbi.nlm.nih.gov/Blast.cgi; accessed on 14 October 2019). Samples were determined to be positive if a band of the expected size was visible in the agarose gel and the sequence yielded a high degree of identity to herpes viral sequences deposited in GenBank.

The electropherograms were visualized using the SeqManPro software of the DNAstar genomic suite package (DNASTAR Inc., Madison, WI, USA) and checked for double peaks and superimposed sequences providing an indication for coinfections. Superimposed electropherograms were separated into major and minor sequences using the Sanger Separator online tool (http://msr.cs.nthu.edu.tw/, accessed on 14 October 2019) and manual inspection.

### 2.3. Statistical Analysis

A multivariable model was used to determine the effect of age, breed, sex, stable and canton on the shedding of each eGHV. Statistical analysis was performed using SPSS. Results were considered significant if *p* < 0.05. Descriptive statistics were used for the remainder of the data.

## 3. Results

### 3.1. Animals

The study population consisted of 18 geldings, 16 stallions, and 34 mares. Horses were aged between 2–32 years. Breeds included Franches Montagnes horses (*n* = 21), Warmbloods (*n* = 12), Ponies (*n* = 11), Trotters (*n* = 7), Arabians (*n* = 5), Thoroughbreds (*n* = 3), Quarter Horses related breeds (*n* = 5) and Frisian (*n* = 1). For three horses, age and breed were unknown.

The nasal swabs originated from horses without overt clinical signs from 14 stables in six Swiss cantons and two stables in Southern Germany close to the Swiss border (Figure 2).

### 3.2. Nasal Shedding of Equid Gammaherpesviruses

eGHV were detected in 40 out of 68 samples (59%); 26 (38%) of these were positive for EHV-2, 8 (12%) for EHV-5 and 6 (9%) for AHV-5 (Figure 3).

eGHVs were detected in 11 out of 14 (79%) stables. In five stables, every examined horse was positive for at least one eGHV. These stables were located in Thurgau, Uri, Vaud, and Germany. The positive number of horses per stable was as follows: TG, *n* = 4; UR1, *n* = 5; UR2, *n* = 1; VD2, *n* = 10, DE1, *n* = 1. No eGHV positive result was obtained in two stables in canton Aargau and one stable in canton Zurich. EHV-2 was present in every stable where at least one eGHV positive result was obtained. EHV-5 was detected in six (43%) stables and AHV-5 was found in three (21%) stables. Aargau was the only canton in which eGHV was not detected at all. In Thurgau and Uri, all tested horses were positive for at least one eGHV.

The odds of shedding EHV-2 decreased with age (*p* = 0.01). For each year’s increase of age, the odds of shedding EHV-2 decreased by a factor of 0.9. For EHV-5 there was no significant association with age. The odds of shedding AHV-5 increased with age (*p* = 0.04). For each increasing year of age, the odds of shedding AHV-5 increased by a factor of 1.1. Breed, sex, canton, or stable showed no significant association with virus shedding for any of the three viruses.

Co-infection with two or more eGHV was observed in 10 out of 40 (25%) positive horses from six out of 12 (50%) positive stables (Table 1). The combination of EHV-5 and EHV-2 was the most frequent co-infection (six cases). EHV-5 was the major and EHV-2 the minor sequences in these cases.

## 4. Discussion

Screening of 68 healthy adult horses from Switzerland for herpesviruses showed shedding of eGHV in over 50% of the animals. Most frequently EHV-2 (38%) was shed. This correlates well with other studies around the world. In New Zealand, the EHV-2 prevalence was estimated at 32% in healthy horses as well as horses with respiratory disease [21]. A recent prevalence study of herpesviruses including 500 healthy Polish horses, reported a 77% EHV-2 prevalence in respiratory tract samples [23]. Lower EHV-2 prevalences in healthy horses were reported from Ethiopia (7%) [11] and Turkey (24%) [22]. Investigations on the clinical relevance of eGHV suggest that EHV-2 may induce or predispose equids to respiratory diseases [11]. Horses shedding EHV-2 were three times more likely to display clinical respiratory disease compared to non-shedders and EHV-2 was found more often in horses with lower airway inflammation compared to healthy horses in several studies [11,21,32].

The second most frequently shed eGHV in our study was EHV-5 (12%). Reported prevalences of EHV-5 in nasal swabs seem generally somewhat higher, ranging from 14–92% [11,20,21,24,27] depending on geographic location, age, and the investigated specimens. In our study, only healthy horses were sampled which may contribute to the relatively low prevalence, while in other studies often also horses with clinical signs were included. However, the aforementioned prevalence study from Poland in which only healthy horses were sampled, also reported a higher prevalence of 47% [23]. These differences highlight the importance of local studies to determine the EHV-5 prevalence. The clinical significance of EHV-5 in adult horses is unclear. EHV-5 was found more often in horses with lower airway inflammation compared to healthy horses [11,21,32] and EHV-5 is suggested as the causative agent of EMPF among other contributing factors that are largely unknown [14,15,17].

Asinine herpesviruses (AHV-4, AHV-5, AHV-6) circulate in donkeys as well as horses [33,34]. The shedding rate of 9% in our study agrees with a prevalence study in Austria in which PBMCs, and nasal and conjunctival swabs of 266 healthy Lipizzaner horses were analyzed. In that study 22 nasal swabs were positive for AHV-5 (8.2%) [34]. There was no evidence of the previous contact with donkeys and no statistically significant association with age, sex, or geographical location [34]. The clinical significance of this virus is unclear. The asinine herpesviruses AHV-4, AHV-5, and AHV-6 were first detected in donkeys with interstitial pneumonia similar to EMPF [35,36]. AHV-5 was detected in horses with respiratory disease, poor performance [12] as well as in two cases of EMPF both of which had a concurrent EHV-5 infection [16,37].

A positive EHV-2, EHV-5, or AHV-5 test result in a horse with the respiratory disease must be interpreted with caution as the shedding rates of 38%, 12%, and 9% respectively, in healthy horses in Switzerland indicate that these viruses readily circulate in the equine population and might be an incidental finding. Similarly, biosecurity measures should not be implemented solely based on a positive test result, instead, multiple factors such as epidemiological factors, herd characteristics, case history, and clinical signs as well as results from other diagnostic tests should be considered.

The decreasing odds of EHV-2 shedding with increasing age can be explained by the epidemiology of the virus. Horses are infected at young ages and acquire immunity over the course of their life, which results in the immune system being able to cope with the virus [27,38,39]. The lack of significant association of age with EHV-5 could be due to the low number of cases or a different infection pattern. Similarly, the shedding rate of AHV-5 was lower than EHV-2 in our study, indicating that this virus is not circulating as frequently, and therefore horses are more likely to be exposed later in life resulting in increased shedding rates with age.

We did not detect any alphaherpesviruses in the nasal swabs by nested consensus PCR. This method targets a highly conserved region of the herpes viral DNA polymerase gene and was previously shown to detect EHV-1 and 4 [30]. Interestingly, some of the samples had previously been included in an EHV-1 and 4 study, and four of them were found weak EHV-4 positive (material intended for publication) [28]) using a specific real-time PCR [40]. However, the sensitivity of broad-range PCR is generally considered lower than that of specific PCR, particularly real-time PCR, and it is therefore not surprising that these weak positive samples were missed by the panherpes PCR. This also indicates that the prevalence of eGHV may be higher if specific real-time PCRs were used. However, an advantage of the panherpes PCR is the fact that co-infection with several herpesviruses may be recognized. Superimposed electropherograms showing double peaks are indications for co-infections and may be separated in silico or by cloning into the single sequences [41] (Figure 4).

Coinfections of several eGHV, predominately of EHV-2 and EHV-5, were seen in approximately 25% of the positive samples. EHV-5 was always the major sequence when EHV-2 was the coinfecting virus. Coinfection of EHV-2 and AHV-5 was less frequently detected, but both viruses were present as major and minor sequences. AHV-5 and EHV-5 coinfection was rarely detected.

These results agree with prior investigations around the world. In Turkey, coinfection with EHV-2 and EHV-5 occurred in 29% of examined nasal swabs [22]. In New Zealand coinfection with multiple herpesviruses (EHV-1, -2, -4, -5) was shown in more than 50% of the virus-positive horses [21]. Also in Poland, coinfection with EHV-2 and EHV-5 was present in 50% of nasal swabs. Detection rates of co-infection vary depending on the sampling site. A considerably lower coinfection rate of 5% in bronchoalveolar fluid (BALF) samples of horses with respiratory disease were reported in a recent study in Ethiopia [11]. However, BALF is a good clinical sample to estimate EHV-5 virus load in the lungs [42] and was strongly associated with EMPF (sensitivity 91%, specificity 98.3%) [43]. In contrast, examination of nasal secretion showed the highest sensitivity (72.7%) and specificity (83.1%) at a level of >254,890 glycoprotein B target genes/million cells to support a diagnosis of EMPF. Nevertheless, a combination of whole blood and nasal secretion can be regarded as clinically useful in support of EMPF diagnosis (sensitivity 90%, specificity 89.8%) [43]. Coinfection rates in PBMC with EVH-2 and EHV-5 of 1% are reported in Australian foals [8]. However, the frequency of co-infection and the number of contributing viruses depend largely on the method used for discrimination. In contrast to using a set of highly specific PCRs, the in silico separation of superimposed electropherograms obtained from the panherpes PCR has the limitation that sequences present in low copy numbers or co-infections of more than two viruses may not be recognized or cannot be separated. Therefore, we cannot exclude that more or other types of co-infections were present in our sample material.

## 5. Conclusions

Although the sampled population is not representative of the entire Swiss horse population, a first insight into the diversity and occurrence of eGHV in Switzerland was obtained. It appears that horses in Switzerland shed a similar range of eGHV compared to reports from other countries. Also, the prevalence of shedding appears similar to other countries.

As EHV-2 can often be found in nasal swabs of healthy Swiss horses, a positive test result may be an incidental finding or at best a contributing factor to a clinical manifestation. EHV-5 and AHV-5 are less frequently diagnosed and there are more reports that associate them with disease, such as EMPF. Therefore, although the frequency of virus detection does not correlate directly with clinical significance, a positive EHV-5 or AHV-5 result should not be underestimated. In any case, biosafety measurements such as quarantine or isolation should not be based on a positive test result alone. Further investigations including a larger sample size as well as a longitudinal sampling of individual horses would be helpful to shed more light on the epidemiology of eGHV in Swiss horses.

## Figures and Tables

**Figure 1 viruses-13-01686-f001:**
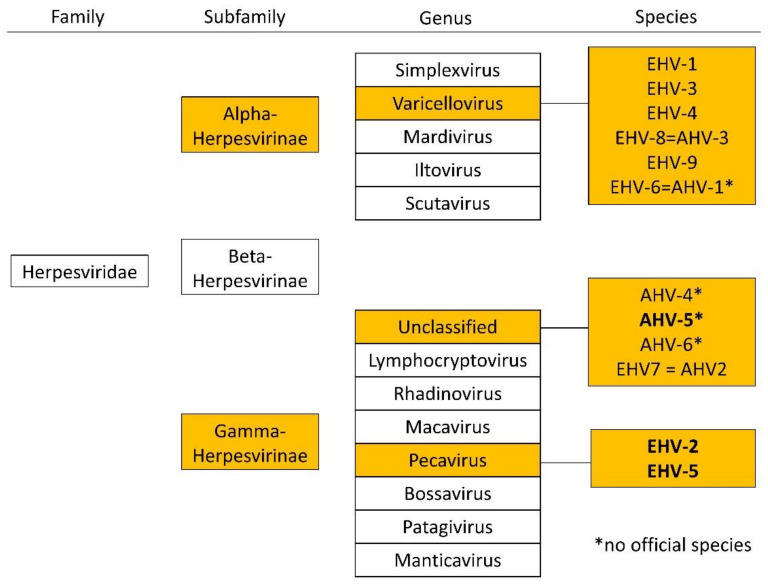
Overview of the family of *Herpesviridae* and relevant subfamilies, genera, and species in horses according to the international committee on the taxonomy of viruses (ICTV), status July 2021. Equid gammaherpesviruses detected in this study are highlighted in bold.

**Figure 2 viruses-13-01686-f002:**
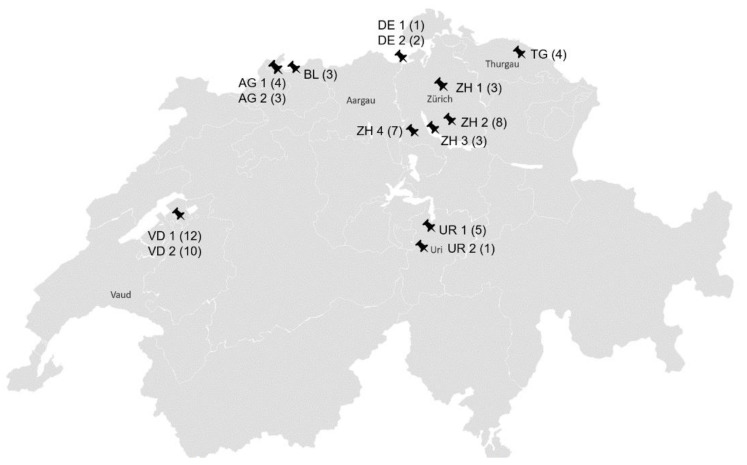
Geographical distribution of the stables; AG: Aargau; BL: Baselland; DE: Germany; TG: Thurgau; UR: Uri; VD: Waadt; ZH: Zurich; ZH 1 (3): canton, stable number, number of sampled horses per stable.

**Figure 3 viruses-13-01686-f003:**
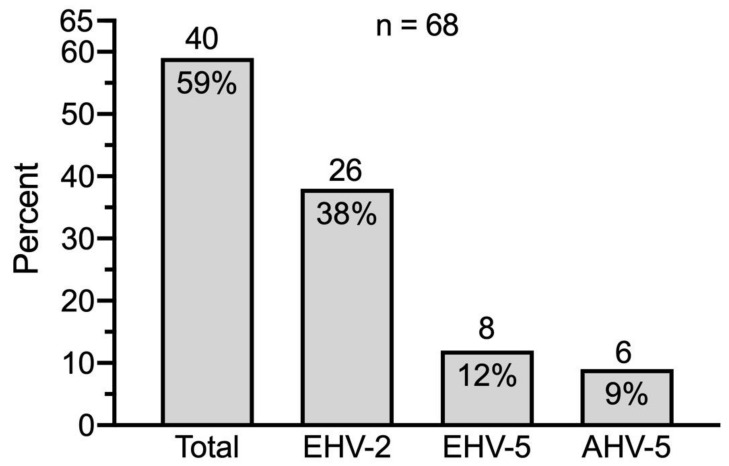
Prevalence of eGHV in nasal swabs of healthy Swiss horses (*n* = 68). Absolute numbers of horses are indicated above the columns.

**Figure 4 viruses-13-01686-f004:**
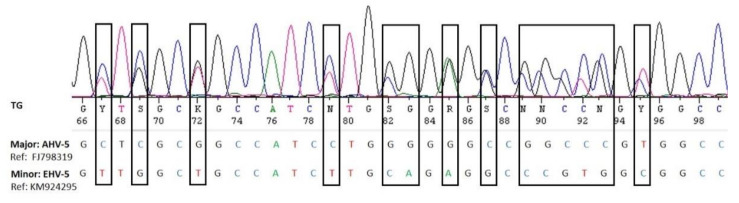
Visualization of the superimposed electropherograms of AHV-5 (major sequence) and EHV-5 (minor sequence) in sample LS030-2 (Table 1).

**Table 1 viruses-13-01686-t001:** Equid gamma herpesvirus coinfections as determined by in silico separation of superimposed electropherograms (Figure 4).

Sample	Stable	Major Sequence	Minor Sequence
LS030-2	TG	AHV-5	EHV-5
LS030-5	UR1	EHV-5	EHV-2
LS030-8	ZH2	EHV-5	EHV-2
LS030-10	ZH4	EHV-5	EHV-2
LS030-11	ZH4	EHV-5	EHV-2
LS033-06	VD1	EHV-5	EHV-2
LS036-07	VD2	EHV-5	EHV-2
LS033-09	VD2	EHV-2	AHV-5
LS036-10	VD1	AHV-5	EHV-2
LS036-11	VD1	AHV-5	EHV-2

TG: Thurgau; UR: Uri; VD: Waadt; ZH: Zurich.

## Data Availability

Not applicable.

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
