# Peer review of "Prevalence of Nasal Shedding of Equid Gammaherpesviruses in Healthy Swiss Horses"

_viruses, 2021, doi:10.3390/v13091686_

Round 1
Reviewer 1 Report
The authors present a straightforward and easy-to-read study on the prevalence of various equine herpesviruses. While prevalence studies have been preformed previously, this is the first to specifically look at horses in Switzerland. Only healthy horses were examined to determine a "background level" of asymptomatic infection and virus shedding. The authors suggest that the data presented here can be used to generate health and safety measures for horses in Swiss stables.
One question I was left with after reading the paper was, is the lack of EHV-1 and EHV-4 in nasal secretions unique to Swiss horses? Is this finding of clinical significance? What is the prevalence of these viruses in other countries? Page 7, Line 255 briefly mentions co-infections between EHV-1, -2, -4, and -5 in New Zealand but no other information is given. Also, given that these viruses were tested for but not found, should the title of the manuscript be changed to "herpesviruses" instead of "gammaherpesviruses?" Zero percent prevalence is still a prevalence.
Minor comments:
1) First sentence of abstract: The word "sings" should be "signs."
2) The authors need to add a sentence or two to the abstract as to why their findings are important to the field.
3) Fig. 1: Highlight the specific viruses that were tested for in this study.
4) Page 4, Line 145: "The nasal swabs originated from healthy horses from 14 stables in six Swiss cantons as well as from two stables close to the Swiss border in Southern Germany." Were there any symptomatic horses in these stables that were excluded from the study?
5) Fig. 3: Add EHV-1 and EHV-4 to the bar graph as 0% to show that these were tested for but not found.
6) Page 6, Line 190: Remove the comma after the word "suggest."
Author Response
We thank the reviewers for their valuable work, helpful comments and suggestions and hope to answer all question and suggestions satisfactorily. Besides of the specific corrections specified below, we have generally revised the English spelling of the manuscript. Line numbers refer to the manuscript with accepted changes.
Reviewer 1 Gammaherpesvirus Scheurer et al Viruses
The authors present a straightforward and easy-to-read study on the prevalence of various equine herpesviruses. While prevalence studies have been performed previously, this is the first to specifically look at horses in Switzerland. Only healthy horses were examined to determine a "background level" of asymptomatic infection and virus shedding. The authors suggest that the data presented here can be used to generate health and safety measures for horses in Swiss stables.
One question I was left with after reading the paper was, is the lack of EHV-1 and EHV-4 in nasal secretions unique to Swiss horses? Is this finding of clinical significance? What is the prevalence of these viruses in other countries? Page 7, Line 255 briefly mentions co-infections between EHV-1, -2, -4, and -5 in New Zealand but no other information is given. Also, given that these viruses were tested for but not found, should the title of the manuscript be changed to "herpesviruses" instead of "gammaherpesviruses?" Zero percent prevalence is still a prevalence.
Authors answers: As mentioned in lines 237-238, 4 of the 68 horses included in this study were found weak positive for EHV-4 in a previous, yet unpublished EHV-1 and -4 study using a highly sensitive diagnostic test (specific EHV-1/4 qPCR). These 4 samples were (not unexpectedly) missed by the panherpes PCR that is known to have a lower sensitivity and that was used in this study (lines 239-240). The data from the EHV1/4 study were not included in this publication because the aim here was to gain information on the diversity and prevalence of GHV in healthy animals while the EHV-1/4 study also included animals from clinical outbreaks. For your personal information (confidential): The EHV1/4 study (n=277) showed a low prevalence of EHV-1 shedding (<1%) while EHV-4 shedding was somewhat more common (approx. 10%). We have rephrased lines 236-241 in the hope to clarify the situation.
Minor comments:
1) First sentence of abstract: The word "sings" should be "signs."
Authors answers: This was addressed as suggested (line 11)
2) The authors need to add a sentence or two to the abstract as to why their findings are important to the field.
Authors answers: This was addressed as suggested. The following sentences were introduced in the abstract:
“The prevalence in healthy Swiss horses is unknow to date but this data would provide valuable information for causal diagnosis in clinical cases and formulation of biosecurity recommendations. As EHV-2 shedding was common in healthy horses a positive PCR result must be interpreted with caution regarding the formulation of biosecurity recommendations and causal diagnosis. As EHV-5 and AHV-5 shedding was less common than EHV-2, a positive test result is more likely to be of clinical relevance.”
At the end of the introduction, we added: “Data on nasal shedding in healthy horses in Switzerland may therefore help with recommendations regarding biosecurity measures and causal diagnosis of clinical cases.”
3) Fig. 1: Highlight the specific viruses that were tested for in this study.
Authors answers: We used a panherpes PCR and therefore did not test for any specific virus species but rather for all herpesviruses. We can therefore not highlight the viruses we tested for but have instead highlighted the viruses detected in the study in bold. We hope this is satisfactory for the reviewer.
4) Page 4, Line 145: "The nasal swabs originated from healthy horses from 14 stables in six Swiss cantons as well as from two stables close to the Swiss border in Southern Germany." Were there any symptomatic horses in these stables that were excluded from the study?
Authors answers: All examined nasal swabs were selected randomly from an archived collection of 200 nasal swabs of horses without overt clinical signs. However, sampling was limited to horses where owners consent was available and therefore not all horses in the stables were clinically examined. We cannot fully exclude, that other horses on the premises have shown mild clinical signs at the sampling timepoint. We knew from the two stables in Southern Germany that an EHV-1 outbreak had occurred a few weeks prior to sampling.
5) Fig. 3: Add EHV-1 and EHV-4 to the bar graph as 0% to show that these were tested for but not found.
Authors answers: See answer to major comment above. The aim of the study was to gain information on the diversity and prevalence of GHV. The panherpesvirus PCR used in this study is not the best available method for EHV-1 and 4 detection. In contrast, for detection of the entirety of GHV, there is no other option than the panherpes PCR. We have addressed the issue in the discussion (line 234-244). We would therefore prefer not to include EHV-1 and -4 in the graph. We hope this is satisfactory for the reviewer.
6) Page 6, Line 190: Remove the comma after the word "suggest."
Authors answers: This was addressed as suggested (line 192).
Reviewer 2 Report
The manuscript by Scheurer et al., entitled "Prevalence of nasal shedding of equine gammaherpesviruses in healthy Swiss horses" performs an study of 68 nasal samples from healthy horses belonging to different geographically dispersed Swiss and German stables to assess the prevalence of gammaherpesviruses in these samples, as well as the possible association with age or location.
The work is well conducted and easy to follow. The study has been well very designed and data are clearly presented.
I have only some minor points to be clarified. In my opinion the main drawback is that the references are old. Only a few are less than 5 years old.
Other minor details: please check spelling of sign in line 11, and add city in line 88.
Author Response
We thank the reviewers for their valuable work, helpful comments and suggestions and hope to answer all question and suggestions satisfactorily. Besides of the specific corrections specified below, we have generally revised the English spelling of the manuscript.
Reviewer 2 Gammaherpesvirus Scheurer et al Viruses
The manuscript by Scheurer et al., entitled "Prevalence of nasal shedding of equine gammaherpesviruses in healthy Swiss horses" performs a study of 68 nasal samples from healthy horses belonging to different geographically dispersed Swiss and German stables to assess the prevalence of gammaherpesviruses in these samples, as well as the possible association with age or location.
The work is well conducted and easy to follow. The study has been well very designed and data are clearly presented.
I have only some minor points to be clarified. In my opinion the main drawback is that the references are old. Only a few are less than 5 years old.
Authors comment: The references were reviewed and new references added where appropriate
Other minor details: please check spelling of sign in line 11, and add city in line 88.
Authors comment: this was addressed as suggested, the name of the city has been added.
Round 2
Reviewer 1 Report
All comments have been addressed adequately.
